# Cardiovascular Complications Associated with COVID-19 and Potential Therapeutic Strategies

**DOI:** 10.3390/ijms21186790

**Published:** 2020-09-16

**Authors:** Arun Samidurai, Anindita Das

**Affiliations:** Division of Cardiology, Pauley Heart Center, Internal Medicine, Virginia Commonwealth University, Richmond, VA 23298, USA; arun.samidurai@vcuhealth.org

**Keywords:** cardiovascular diseases, COVID-19, treatment strategies

## Abstract

The outbreak of coronavirus disease 2019 (COVID-19), an infectious disease with severe acute respiratory syndrome, has now become a worldwide pandemic. Despite the respiratory complication, COVID-19 is also associated with significant multiple organ dysfunction, including severe cardiac impairment. Emerging evidence reveals a direct interplay between COVID-19 and dire cardiovascular complications, including myocardial injury, heart failure, heart attack, myocarditis, arrhythmias as well as blood clots, which are accompanied with elevated risk and adverse outcome among infected patients, even sudden death. The proposed pathophysiological mechanisms of myocardial impairment include invasion of SARS-CoV-2 virus via angiotensin-converting enzyme 2 to cardiovascular cells/tissue, which leads to endothelial inflammation and dysfunction, de-stabilization of vulnerable atherosclerotic plaques, stent thrombosis, cardiac stress due to diminish oxygen supply and cardiac muscle damage, and myocardial infarction. Several promising therapeutics are under investigation to the overall prognosis of COVID-19 patients with high risk of cardiovascular impairment, nevertheless to date, none have shown proven clinical efficacy. In this comprehensive review, we aimed to highlight the current integrated therapeutic approaches for COVID-19 and we summarized the potential therapeutic options, currently under clinical trials, with their mechanisms of action and associated adverse cardiac events in highly infectious COVID-19 patients.

## 1. Introduction

Coronavirus-19 (COVID-19) is an emerging infectious disease caused by the novel single-stranded RNA enveloped Severe Acute Respiratory Syndrome-Coronavirus-2 (SARS-CoV-2). The first case of COVID-19 was reported on 8 December 2019 in Hubei province of China [1] and within a short span of time, the disease quickly spread to other parts of the world [2] and has rapidly evolved as a global pandemic situation. The first confirmed case of COVID-19 in the United States of America (USA) was reported on 20 January 2020 in the state of Washington when a 35-year-old man showed symptoms of SARS-CoV-2 infection after returning from Wuhan, China [3]. The first person-to-person transmission of a confirmed COVID-19 case in USA was reported in Illinois on 30 January 2020 after an initial positive diagnosis of COVID-19 on the patient’s wife, who returned from Wuhan, China in mid-January 2020, and unfortunately, COVID-19 is now widespread in all 50 states across the USA [4,5]. According to Johns Hopkins Coronavirus Resource Center, as of 9 September 2020, there are 27,699,974 confirmed COVID-19 cases and 900,239 confirmed deaths worldwide [6]. USA is now the epicenter of the disease; the death toll has reached 190,763 with 6,358,983 confirmed COVID-19 cases [7]. Figure 1 is the graphical representation of most affected regions of confirmed cases with reported deaths across the world. Countries with more than 100,000 confirmed cases with reported deaths are presented in Figure 1A,B; and countries with more than 50,000 cases (but fewer than 100,000) and corresponding reported deaths are shown in Figure 1C,D.

## 2. Structure and Genomic Organization of Novel SARS-CoV-2

Understanding the structure and the genetic makeup of SARS-CoV-2 is important to appreciate the ongoing efforts to address this disease and for the discovery of drugs and vaccines. SARS-CoV-2 is spherical in shape and consists of multiple components which are essential for their replication and transcription: (1) Several club shaped projections on the surface of the envelope, called spike glycoprotein (S), which helps in anchoring to the host cell and acts as an inducer to neutralize antibodies, (2) A small membrane envelope protein (E), (3) Structural membrane protein (M), which spans the lipid bilayer, (4) Hemagglutinin-esterase glycoprotein (HE), which destroys the sialic acid present on the host cell and helps the virus to inject its genetic material, (5) Nucleoprotein (N) and (6) the key component, the positive-sense single-stranded Genomic RNA [8,9,10]. The typical structure of COVID-19 virus depicting the above-mentioned components is shown in Figure 2A. 

The genome size of COVID-19 is about 26.4–31.7 kb [11] and is the largest among all known RNA viruses in this category (SARS-CoV and Middle East respiratory syndrome, MERS). SARS-CoV-2 encompasses several open reading frames (ORF) along with its 5′ UTR and 3′ UTR regions. ORF1a/b (frame shift) is the full-length gene (29.8 Kb size) that encodes replicase polyprotein named pp1a protein and 16 accessary (non-structural proteins (nsps) and ORF1b codes for pp1b and 10 nsps [11,12,13,14]. The structural proteins including spike (S), envelope (E), membrane protein (M) and nucleoprotein (N) are coded by ORFs 10 and 11 present on the 3’ UTR region. Several other essential accessary proteins are coded by ORF3, ORF7a, ORF7b, and ORF8 [15]. The domain structure of SARS-CoV-2 is presented in Figure 2B.

The genomic sequencing data obtained from COVID-19-infected patients in China revealed the distinct features of SARS-CoV-2 [16]. Sequence comparison showed the novel SARS-CoV-2 was more distantly correlated with SARS-CoV (about 79%) and MERS-CoV (about 50%). Some of the salient features of SARS-CoV-2 make it unique and virulent compared to previously known coronaviruses. Reports suggest that SARS-CoV-2 lacks the ORF8a protein present in SARS-CoV and has variation in the 3c and 8b proteins [12]. In addition, a single mutation at N501T in the spike protein strengthened the binding efficiency of SARS-CoV-2 to Angiotensin-Converting Enzyme 2 (ACE2), the primary mode of entry into host cell [17]. Structural details of SARS-CoV binding to the host cell through spike protein give us the clue about the importance of the mutation in this region [18]. Studies show that this mutation in the spike protein of SARS-CoV-2 increases its binding affinity to ACE2 in human by 10–20 times higher than SARS-CoV [19]. This mutation in the spike protein may be one of the key attributes of SARS-CoV-2 that led to its rapid spreading around the world in a very short period.

There are six different strains of SARS-CoV-2 identified so far, namely, (1) L strain (originated in Wuhan, China and the parent ORF8-L84S Strain), (2) S strain (mutation of ORF8, L84S), (3) V strain (variant of ORF3a coding protein NS3, G251V), (4) G (mutation in Spike protein, D614G), (5) GH strain (mutations in Spike protein, D614G and ORF3a, Q57H) and (6) GR (mutation in nucleocapsid gene, RG203KR) [20,21]. Among these variants, G strain is the most widespread, has undergone several mutations since January 2020, and branched into subtypes GR and GH. The G and GR strains are prevalent in Europe, including in Italy, whereas the GH strain is widespread in Germany and France. The most predominant variant found in North America is the GH strain, and this subtype along with the GR clades accounts for 74% of all global sequences of SARS-CoV-2 genome [22]. As of now, strains G, GH and GR are constantly increasing, globally, and it is yet to be determined whether the unique nature of these strains is associated with the disease intensity.

## 3. Mode of Entry into the Host Cell

COVID-19 virus is predominantly transmitted from human to human through respiratory droplets or aerosols (>5–10 μm in diameter) and contact routes. Inhalation of respiratory droplets and aerosols from COVID-19-infected persons is the most likely potential mode of transmission of the disease [23]. Once in the host system, SARS-CoV-2 follows the traditional steps similar to any other virus for its mode of entry into the host cells [24]. The spike protein anchors the virus to the surface of the host cell by binding to the ACE2 receptor [25]. The virus undergoes conformational changes to fuse to the cell membrane of the host cells and engulfs into the cytoplasm of the cell by endosomal pathway. Once inside the cell, the virus releases its genomic RNA and multiplies using the host’s molecular machinery. Experimental evidence using HeLa cells demonstrate that SARS-CoV-2 entry into host cell is activated by cell surface proteases and lysosomal proteases such as Transmembrane Serine Protease 2 (TMPRSS2) and lysosomal cathepsin [26]. ACE2 is expressed in type II alveolar cells, the predominant portal of entry in the lungs; it is also expressed in the heart, intestine and kidney as well as on the epithelial cells of oral mucosa and the tongue [27,28]. SARS-CoV-2 primarily affects the respiratory tract, and infected patients suffer from pneumonia and flu-like symptoms (Figure 3). The patients might need intensive care and artificial ventilation after developing acute respiratory syndrome (ARDS) or multiple organ dysfunction syndrome (MODS).

## 4. Pulmonary Diseases

The lungs, being a major organ targeted by SARS-Cov-2 infection, are severely compromised in delivering their function. One of the common clinical manifestations of COVID-19 at the late stage of the disease is shortness of breath, pneumonia-like symptoms and hypoxia, which ultimately is fatal to the patients. In the pulmonary vasculature, SARS-CoV-2 enters through endocytosis and activates ADAM metallopeptidase domain 17 (ADAM17), which in turn cleaves ACE2, which indicates the loss of protection against the renin–angiotensin–aldosterone system (RAAS), which is mediated by cleaved ACE2 [29,30]. The activation of ADAM17 also triggers acute pulmonary inflammation and infiltration of cytokines and leukocytes in the alveolar space and results in pulmonary edema [9,31].

Overactive systemic inflammation as a response to COVID-19 infection results in cytokine storm and leads to respiratory difficulties and accounts for majority of the deaths during end stage of the treatment [32]. Pulmonary complications associated with COVID-19 infection include diseases such as acute respiratory distress syndrome (ARDS), vascular endothelialitis, sepsis, pulmonary edema and pulmonary embolism [33,34]. A multicenter cohort study involving 235 hospitals from 24 countries, which includes 1128 patients and 294 cases of confirmed COVID-19, suggests that up to 51.2% suffer severe pulmonary complications post-surgery and the majority of the deaths are largely due to pulmonary embolism [33]. Lung autopsy reports obtained from COVID-19 patients who died due to ARDS showed severe alveolar damage and infiltration of perivascular T-cells. Histology analysis also demonstrated increased thrombus formation, intussusceptive angiogenesis and microangiopathy in COVID-19 patients compared to influenza [32]. Gene expression analysis using RNA isolated from COVID-19 patients showed several inflammatory markers and angiogenesis-related genes were differently regulated compared to healthy lungs. Most importantly, there was a significantly increased positive count for ACE2 in COVID-19 tissues compared to control. COVID-19 patients with ARDS were also characterized by an increased deposition of fibrin and high expression of D-dimers and fibrinogen, suggesting fibrinolysis as a determining factor of mortality [35]. Although pulmonary complication is the dominant clinical manifestation of COVID-19, underlying cardiovascular complication as well as developed acute cardiac injury enhances the vulnerability of the patient. Acute respiratory complication/failure and cytokine storm may cause reduced oxygen supply, which could lead to acute myocardial injury in COVID-19 patients [36].

## 5. Cardiovascular Complications of COVID-19

Patients with cardiovascular disease (CVD) have an increased risk for severity and mortality with COVID-19 infection, mainly because of the abundance of ACE2 receptor in the cardiovascular system, which serves as a gateway for the entry of virus in lungs and heart [37]. Respiratory illness and acute cardiac injury are major clinical manifestations observed in patients infected with SARS-CoV-2 during the late stage complications of the disease [38]. Reasonably, patients with coronary artery disease or heart failure are vulnerable to developing major cardiac injury, and once such patients are infected with SARS-CoV-2, they are at greatest risk of serious myocardial impairment or cardiac dysfunction, requiring hospitalization due to unexpected deterioration, and eventual mortality is greater among these patients. A brief view of cardiovascular complications associated with COVID-19 is presented in Figure 3. 

Voluminous available clinical evidence confirms that the severity of COVID-19 is pronounced in patients with a prevalence of underlying cardiovascular diseases, and in many of these patients, the virus causes severe myocardial injury [39], including myocardial dysfunction, cardiomyopathy, arrhythmias and heart failure during the course of critical illness [40,41,42,43,44,45,46]. According to the death data of the CDC (Centers for Disease Control and Prevention), different health conditions contribute to the deaths of COVID-19 patients in United States, which are summarized in Figure 4 [47]. Deaths that are associated with more than one underlying condition, e.g., deaths involving both diabetes and respiratory arrest or cardiovascular complications and respiratory arrest, etc. Hypertension, diabetes, cardiac arrest, ischemic heart disease, and heart failure are the major risk factors and have contributed to the fatalities in COVID-19 cases.

The renin–angiotensin–aldosterone system (RAAS) consists of an enzymatic cascade that controls blood pressure by regulating circulatory homeostasis, body fluid and systemic vascular resistance, all of which are involved in the regulation of a myriad of cardiovascular system [48]. ACE1 (angiotensin I-converting enzyme) cleaves angiotensin I (Ang-I) to angiotensin II (Ang-II), which binds to and activates Angiotensin Type 1 Receptor (AT1R), which leads to vasoconstriction, inflammation, fibrosis and proliferation [28] (Figure 3). ACE2 converts Ang-II into angiotensin 1-7 (Ang 1-7), which has vasodilating and anti-inflammatory effects by binding to MAS receptor (MAS-R). ACE2 also cleaves Ang-I into angiotensin-1–9, which is further converted into Ang 1–7 by ACE. Therefore, ACE2 regulates abnormal activation of the RAAS, which can prevent the development of hypertension, cardiac hypertrophy, and heart failure [49]. An increase in ACE2/ACE1 ratio protects against endothelial dysfunctions and vascular constriction, and exogenous ACE2 activation attenuates thrombus formation and reduces platelet attachment to vessels [50,51]. 

The etiology of ACE2-dependent cardiovascular complications with COVID-19 infection is rather complex. SARS-CoV-2 enters cardiovascular cells/tissue by binding to the membrane-bound form of ACE2 (ACE2 receptor). Elevated levels of ACE2 and its activity are the biomarkers of cardiovascular disease including patients with heart failure [52], which indicate that these patients may be more susceptible to COVID-19 infection [53], with worsened prognosis of cardiovascular disease treatment [54]. Measuring plasma angiotensin peptides and plasma ACE2 levels can provide a direct evaluation on the progress of treatment and the state of the RAAS in COVID-19 patients [55]. Nevertheless, earlier clinical studies conveyed that treatment with soluble form of recombinant human ACE2 (rhACE2; APN01 (0.4 mg/kg, IV, BID for 7 days), GSK2586881: 0.4 mg/kg, IV, BID for 3 days) neutralized the excessive SARS-CoV virus in the system and preserved the protective cellular effects of ACE2 in ARDS patients [56,57]. Consequently, scientists propose the therapeutic potential of soluble recombinant ACE2, which can overwhelm SARS-CoV-2 to prevent its binding to cellular ACE2 [58]. In addition, ACE inhibitors (ACEi), which upregulate ACE2 expression on the cell surface, have been proven to be successful, and improved the survival rate in patients undergoing COVID-19 treatment [49]. Abundant expression of ACE2 on the cell surface following virus infection may maintain Ang-II degradation, which could reduce AT1R activation and the risk of deleterious outcomes of COVID-19. 

While the ACE2 gene is located on the X-chromosome, gender has an impact among COVID-19 patients, where men are at increased risk of susceptibility to COVID-19 infection and CVD complications due to their hemizygous allele for ACE2 compared to heterozygous allele in female [28]. Interestingly, clinical data from 1485 European men and 537 women with heart failure showed elevated level of circulating plasma ACE2 in men than in female [59]. This data complements with an observation of increased prevalence and susceptibility of COVID-19 in males and demonstrates that abundance of ACE2 receptor in cardiovascular cells can lead to severe clinical complications [60,61,62]. 

### 5.1. Hypertension

Considering the importance of ACE2 in the development of hypertension and diabetes mellitus, patients with COVID-19 exhibit severe comorbidities including hypertension and diabetes with poor prognosis. Initial evidence from 44,672 confirmed cases in China showed 4.2% with CVD and 12.8% with hypertension. However, among the death rate, 6% had hypertension, 7.3% had diabetes and 6.3% suffered from chronic respiratory disease [62,63]. In another study involving a small population of 99 patients, 40% had underlying CVD or cerebrovascular disease [64]. Interestingly, data from a small registry of 41 admitted patients showed an alarming 73% were men, with a median age of 49 [9]. However, data involving 5700 patients (admitted during 1 March to 4 April 2020) with a median age of 63 from New York City, the epicenter of COVID-19 spread in the USA, showed a slightly different picture [38]. The most common underlying comorbidities were hypertension (57%), obesity (42%), and diabetes (34%). Figure 4 depicts the statistics on distribution of underlying conditions among COVID-19 patients based on the data from Center for Disease Control Department (CDC), USA. The data reveals that hypertension is a major comorbid factor for COVID-19 fatalities.

An increased risk of COVID-19 death was associated with an age greater than 65 years (mortality of 10%), CVDs (coronary artery disease: 10.2%; heart failure: 15.3%; cardiac arrhythmia: 11.5%), chronic obstructive pulmonary disease (14.2%), and current smoking (9.4%). Another detailed observational meta-analysis (49,076 confirmed COVID-19 case) of data available from public domains including databases from MEDLINE, Embase and web of Science showed patients with preexisting condition of CVD, diabetes and hypertension are significantly associated with a higher risk of developing severe complication with COVID-19 disease. Precisely, the analysis comparing the complications between severe vs non-severe (mild to moderate) COVID-19 cases concluded that CVD was significantly associated with increased illness severity and adverse outcomes among COVID-19 patients [65]. Recently, CDC suggests that children with certain medical conditions, like neurological, genetic, and metabolic conditions, or congenital heart disease might be at increased risk of severe illness from COVID-19 compared to other children.

Additional study comprising 191 patients from two hospitals in Wuhan, China reported 48% of patients had underlying comorbidity factors: 30% with hypertension, 19% with diabetes and 8% with coronary heart disease [40]. In another cohort with 1591 confirmed COVID-19 patients (during 20 February and 18 March 2020) with an average median age of 63 from Lombardy, Italy, 68% had at least one underlying comorbidity, hypertension (49%), CVD (21%), hypercholesterolemia (18%), or diabetes (17%) [66]. Moreover, a staggering 82% of the patients were male and the mortality rate was higher in elderly patients aged ≥64 years compared to younger patients (36% vs. 15%). Due to the high prevalence of hypertension in the older population, elderly male individuals may be at the highest risk of infection with worse outcomes, and an increased mortality rate with respect to younger patients.

Patients with hypertension are mostly treated with ACE inhibitors (ACEi) and angiotensin II type-I receptor blockers (ARBs), which substantially increases the expression of ACE2, due to negative feedback activation caused by low level of Ang-I in the system [67]. Considering ACE2 as a preferential receptor of SARS-CoV-2, the patient with antihypertensive therapy with ACEi/ARBs may be at higher risk of developing severe COVID-19 with poor prognosis [67]. Remarkably, clinical studies do not support this hypothesis and found no evidence to demonstrate use of ACEi or ARB as a risk factor in COVID-19 patients [49]. Multiple investigators have demonstrated the beneficial therapeutic effect of ACEi or ARB in prevention of COVID-19 infection [68,69]. Independent studies conducted among hypertensive patients found no association between the use of ACEi or ARB and increased risk of mortality in COVID-19-positive cases [70,71]. A population-based case–control study in the Lombardy region of Italy with a total of 6272 patients with COVID-19 (21 February and 11 March 2020) reported that 22.2% patients were receiving ARB and 23.9% patients were receiving ACEi [70]. Other antihypertensive drugs were also used more in COVID-19 patients than in controls and they had a more frequent history of hospitalization due to cardiovascular complications. However, this study showed no evidence of association of use of anti-hypertrophic drugs including ACEi or ARBs and susceptibility of COVID-19. Another study with 12,594 patients in the New York University (NYU) Langone Health, in which 5894 patients were positive for COVID-19 (46.8%), reported 4357 patients had a history of hypertension (34.6%) [72]. Among these hypertensive patients, 2573 (59.1%) patients were positive for COVID-19 (59.1%). This study also identified no substantial adverse effect with the use of antihypertensive drugs including ACEi or ARBs in the COVID-19 positive patients. 

Therefore, prospective research is warranted to clarify the accuracy of existing contradictory hypotheses regarding the use of ACEi or ARBs to control of blood pressure in hypertensive patients during viral infections. Fundamentally, after entering into the cells via ACE2 receptors and excessive binding of the SARS-CoV-2 result in the downregulation of ACE2 by intracellular degradation and shedding, which could reduce the Ang-II degradation and activation of AT1R with induction of myocardial hyper-inflammatory reaction in response to COVID-19 [49,73]. Due to ACEi or ARB treatment, more ACE2 may be localized in the cell surface after virus binding, which could facilitate Ang-II degradation with reduction of AT1R activation [49]. 

### 5.2. Myocardial Injury and Heart Failure

Apart from hypertension and age, acute cardiac injury, chronic heart damage and heart failure have all been observed in patients treated for COVID-19 infection [44,74]. Due to acute inflammation, procoagulant stimulus and endothelial cell dysfunction, various influenza RNA viruses are involved in the development of human atherosclerotic plaques and progression of atherosclerosis. De-stabilization of vulnerable atherosclerotic plaques triggers acute myocardial infarction (MI) or cardiovascular death [75,76]. Myocardial infarction, commonly known as heart attack, is a clinical condition, where oxygen supply to the heart is restricted and results in the irreversible loss of cardiomyocytes due to activation of cardiac apoptosis [77,78,79]. A large population of patients diagnosed with COVID-19 has died due to MI [80]. Data obtained from a laboratory in Lombardy, Italy suggest that 60.7% (17 out of 28 cases) of patients with confirmed COVID-19 and an existing condition of ST-elevation myocardial infarction (STEMI) had to undergo a repeated coronary angiogram and coronary lesion was identified as a major cause of the complication [81]. Myocardial injury was also identified as a major contributor of mortality in COVID-19 patients, as derived using data from hospitals in Wuhan, China [39]. Strikingly, the cardiac Troponin I (cTnI) level, a distinct marker of myocardial injury, was noticeably elevated in 52 patients out of 187 hospitalized patients with COVID-19 (27.8%) and the mortality was nearly 70% in these patients with elevated cTnI. Progressive increased levels of C-reactive protein and N-terminal pro-B-type natriuretic peptide (NT-proBNP) coexisted with elevated cTnI levels in these COVID-19 patients, which enhance the severity of inflammation and ventricular dysfunction. 

Among 138 patients treated for COVID-19 (admitted to Zhongnan Hospital of Wuhan University during January 2020), 33 patients had either acute myocardial injury or cardiac arrhythmia, as suggested by their elevated cTnI level of 0.011 ng/mL versus 0.0051 ng/mL for those who were treated in non-ICU [82]. Several other retrospective multi-center cohort studies from China have also confirmed the significant elevation of biomarkers of myocardial injury over the course of COVID-19 infection that were strongly associated with rapid surge of irreversible clinical deterioration and increased mortality [40,41,44,83]. Although limited data are available on the incidence of heart failure in patients with COVID-19, the study with 191 hospitalized patients with confirmed COVID-19 (ranging in age from 18 to 87 years) in Wuhan, China (until 31 January 2020) reported that heart failure was identified in 44 patients (23%), among them, 28 (52%) patients died and 16 (12%) patients recovered [40]. Cardiac injury, as a common complication (19.7%), was associated with an unexpected high risk of mortality during hospitalization of elderly patients with COVID-19 in Wuhan, China [41]. Evidence indicated in another retrospective study that apart from ARDS and sepsis, acute cardiac injury (77%) and heart failure (49%) were the most common critical complications of death in 113 deceased patients with COVID-19 in Wuhan, China [84]. Several other case reports also established that acute or end-stage heart failure was the main pathophysiological manifestation of COVID-19 [61,85,86], which might be associated with hyperinflammation and oxidative stress [53,87,88].

Interestingly, one recent study indicated a decline in emergency department visits for heart failure during the COVID-19 pandemic, partly due to effective remote clinician–patient interactions [89]. Since patients with CVD are considered to be more vulnerable to SARS-CoV-2 infection, with higher risk of negative consequences, these patients avoid frequent hospital visits and prefer alternative remote management. However, analyzing the clinical records during the COVID-19 pandemic (between 20 February and 20 April 2020) of emergency department of San Filippo Neri Hospital in Rome, Italy, a study revealed patients with acute heart failure often reported to the emergency department after significant clinical deterioration with high mortality due to failure of routine clinical assessment [90].

Emerging studies indicate that severe COVID-19-related death is associated with coagulopathy, venous thromboembolism ((VTE) and disseminated intravascular coagulation (DIC) [91]. Data obtained from the COVID-19 patient population in Wuhan, China indicate an abnormal coagulation pattern with prolonged prothrombin time [91]. There were 183 patients registered in this study and parameters such as (DIC), antithrombin activity, prothrombin time (PT) and D-dimer, a fibrin degradation product, were measured and compared between survivors and non-survivors. The results showed elevated levels of DIC and D-dimers and prolonged PT in non-survivors and suggest thrombus formation may have contributed to the mortality in these patients. This notion is strongly supported by the observation that treatment of COVID-19 patients with anti-coagulation drug heparin resulted in reduced mortality rate [92]. The 28-day mortality study between heparin users and nonusers indicated that only selected COVID-19 patients with markedly higher sepsis-induced coagulopathy (SIC) score or elevated D-dimer were benefited from the anticoagulant therapy. Notably, anticoagulant treatment may endanger those patients without significant coagulopathy, because the activation of coagulation with local thrombosis/fibrin deposition could limit the survival and dissemination of microbial pathogens and reduce their invasion [93].

### 5.3. Myocarditis

Myocarditis is a disease marked by the inflammation of the heart muscle, most often due to viral infection. This inflammation interferes with the electrical system and compromises the pumping capacity of the heart and results in arrhythmia and cardiac arrest [94]. Common diagnosis procedures include electrocardiogram (ECG), MRI (magnetic resonance imaging) and a manifestation of increased cardiac Troponin I (cTnI) level. COVID-19 patients with severe stage of illness manifest systemic hyperinflammation syndrome [95]. This data suggests an effect of adverse inflammatory reaction or cytokine storm in response to COVID-19 treatment and defines a strong role for ACE2 signaling in COVID-19 disease [95]. Several reports have shown that patients with COVID-19 infection are diagnosed with myocarditis [41,44,61,66]. In a case report of a 69-year-old man admitted in Lombardy, Italy with respiratory difficulties and required mechanical ventilator, with worsening heart condition. Transthoracic echocardiography showed mild left ventricle hypertrophy (LVH) with preserved left ventricular ejection fraction and normal wall motion and elevated plasma Troponin level (at 9.0 ng/mL) [83,96]. Cardiovascular MRI was suggestive of myocarditis and the patient tested positive for COVID-19 infection demonstrating SARS-CoV-2 infection was the most likely cause for the incidence of myocarditis [96]. Similarly, a 53-year-old healthy woman was diagnosed with acute myopericarditis upon COVID-19 infection. Cardiac MRI showed a severe left ventricular dysfunction (Ejection fraction-35%). The patient also had myocyte necrosis with high-sensitivity cardiac Troponin T (hsTnT) level concentration of 0.24 ng/mL [61]. These reports suggest that patient with COVID-19 infection are prone to myocarditis, and physicians would suspect such conditions along with underlying morbidity factors like hypertension and other CVD.

### 5.4. Myocardial Arrhythmias

Emerging clinical and epidemiological evidence suggests that metabolic disarray, hypoxia and accentuated myocardial inflammation due to SARS-CoV-2 infection plays a critical role in the pathophysiology of myocardial injury and prevalence of arrhythmic complications [97]. In a clinical cohort with 138 patients with COVID-19 in Wuhan, China, cardiac arrhythmias were considered a major complication in 23 patients (16.7%) who were transferred to the intensive care unit (ICU) [44]. Specifically, cardiac arrhythmia was more common in ICU patients than in non-ICU patients (44.4% vs. 6.9%). A recent study from New York-Presbyterian/Columbia University Irving Medical Center highlighted the spectrum of life-threatening arrhythmias observed in four patients with COVID-19 infection [98]. Fulminant myocarditis with cardiogenic shock could also coexist with atrial and ventricular arrhythmias, which could increase the severity of COVID-19 patients, including death [99,100]. Therefore, the expected cardiac arrhythmogenic effect of COVID-19 may be an important underlying risk of disease complication, which needs additional precautions and specialized management.

Based on the available clinical data, potential myocardial injury is a relevant challenge among hospitalized patients with COVID-19 with increased risk of mortality; therefore, it is essential for multidisciplinary assessment, including blood pressure control in hypertensive patients as well as cardiovascular evaluation and therapy to reduce the morality for COVID-19 infection. Strikingly, a recent study in Germany involving 100 patients with an average age of 49 years who recently recovered from COVID-19 infection recognized the cardiovascular sequelae, irrespective of preexisting cardiac conditions [101]. Cardiovascular Magnetic Resonance Imaging (CMR) revealed that 78 patients had abnormal cardiac structural changes, 76 had detectable levels of biomarker of cardiac injury, e.g., elevated level of high-sensitivity cardiac Troponin T (hsTnT), lower left ventricular ejection fraction, higher left ventricle volumes, higher left ventricle mass, and raised native T1 and T2 (quantitative assessments of the myocardium composition), commonly found after a heart attack, and 60 had signs of inflammation.

The exact molecular mechanism by which SARS-CoV-2 virus leads to cardiomyocyte injury is not completely understood. However, the abundant expression of ACE-2 receptors in the heart plays an important role in the accumulation of SARS-CoV-2 virus in the cardiac tissue, which eventually results in hyperactivation of inflammation and cardiac tissue injury in patients. Recently, autopsy results of 39 patients, who died at early stage of COVID-19 infection in Germany, revealed the most likely localization of SARS-CoV-2 not to be in the cardiomyocytes, but in interstitial cells or macrophages invading the myocardial tissue [102]. However, another emerging study using human induced pluripotent stem cell-derived cardiomyocytes (hiPSC-CMs) shows SARS-CoV-2 can directly enter and replicate in hiPSC-CMs and induce apoptosis, which results in cessation of cardiomyocyte beating after 72 h of infection [103].

The majority of the COVID-19 patients suffering from cardiovascular complications show a significant elevation of cTnI, NT-proBNP and interleukin-6 (IL-6) or other cytokines [IL1B, IL1RA, IL7, IL8, IL9, IL10, C-X-C motif chemokine 10 (CXCL10), chemokine (C-C motif) ligand 2 (CCL2), granulocyte-macrophage colony-stimulating factor (GM-CSF), and tumor necrosis factor-α (TNF-α)] in their blood stream [9,40,104]. Severe hyperinflammation or cytokine storm due to immunological dysregulation may be the primary contributor to cardiomyocyte injury [105]. Epidemiological studies with other viral RNAs indicated that after entering into the cytoplasm of cardiomyocytes, viral RNA is further transcribed and translated into the viral structural proteins to form the complete infectious virion [106]. Ultimately, infected cardiomyocytes would be lysed, which could lead to activation of the innate immune response with induction of pro-inflammatory cytokines, inflammation-induced destabilization of coronary artery plaques and development of left ventricular dysfunction [107]. Collectively, uncontrolled hyperactivated T-lymphocytes with systemic inflammation appears to be the most common mechanisms of the cardiomyocyte injury in COVID-19 patients with profound cardiovascular consequences. 

In addition to binding to ACE2 of the host cell, the priming of the transmembrane spike (S) glycoprotein of SARS-CoV-2 by host proteases (furin, a signature protease of highly pathogenic viruses) through cleavage at the S1/S2 and the S2′ sites could enhance its transmissibility and pathogenicity [108]. Multiple evidences have revealed that the Notch signaling plays a major role in maintaining the homeostasis of the cardiovascular system, including atherosclerosis progression and ventricular remodeling after myocardial infarction [109,110,111]. Furin is transcriptionally induced by Notch signaling, but Notch is cleaved at the cell membrane by ADAM10/ADAM17 to enable final cleavage by γ-secretase to form active Notch intracellular domain, which regulates the transcription of target genes in nucleus. Therefore, targeting Notch activation using inhibitor γ-secretase (GSI) could be a promising therapeutic strategy to block the virus entry into the cardiac cells by reducing furin and increase ADAM17 shedding. The Notch signaling also modulates the activity of innate and adaptive immune responses by inducing macrophage polarization [112]. In microphages, it directly binds to IL-6 promoter in response to interferon (IFN)-γ and promotes IL-6 production, which may cause severe myocardial injury due to triggered “cytokine storm” [113]. Our current understanding on the molecular mechanisms of cardiomyocyte injury for SARS-CoV-2 infection is limited and future in depth rigorous studies are warranted.

## 6. Treatment Options and Cardiac Complications Due to COVID-19 Treatment 

Early diagnosis of COVID-19 infection in patients is crucial for the recommendation of appropriate treatment strategy and to address associated CVD complications. Initial symptoms of SARS-CoV-2 infection include high fever or chills, cough, shortness of breath, headache, sore throat, new loss of taste or smell, diarrhea and fatigue that appears during 2–14 days after the exposure to the virus. These early indications, though similar to regular viral infection, should be taken seriously during this pandemic time and diagnosed further for the presence of COVID-19 infection. Currently, the established diagnostic test for the identification of SARS-CoV-2 infection is based on Nucleic acid amplification testing (NAAT) or commonly called real-time reverse transcription- polymerase chain reaction (RT-PCR) assay, nucleic acid-based meta-genomic next-generation sequencing (mNGS), reverse transcription loop-mediated isothermal amplification (RT-LAMP) and antigen testing performed with Nasopharyngeal swab specimen [114,115].

In the absence of any pharmaceutical interventions, traditional public health measures are considered to be the mainstay of management tools to curb this worldwide COVID-19 epidemic. Most widely accepted practices are hygienic precautions, isolation and quarantine, social distancing and community containment [44,116]. To minimize cardiovascular complications in highly infectious COVID-19 patients, the patients with COVID-19 infection require routine monitoring of cardiac parameters with echocardiography, telemetry to assess QT interval and electrocardiograph (ECG) to identify the development of cardiomyopathy, arrhythmia, ischemic heart disease and heart failure.

Potential therapeutic options to impede the propagation of COVID-19 and its associated cardiovascular complication are desperately needed during this ongoing severe pandemic. Researchers and clinicians are focusing on developing new drugs against coronavirus as well as repurposing already approved drugs for the treatment of COVID-19 patients. Unapproved antiviral drugs for SARS-CoV-1 and/or Middle East respiratory syndrome coronavirus (MERS-CoV) diseases are also currently being reevaluated as treatment options for COVID-19. However, COVID-19 poses unique problems that were not encountered with the previous known viruses. The major issue was to address the CVD complications, systemic and vascular inflammation, and to deal with comorbid risk like hypertension, diabetes and heart failure. Initial approaches were to emphasis on obstructing the viral replication and inflammation by using antiviral drugs, such as, Remdesivir, liponovir/ritonavir, hydroxy chloroquine (HCQ), corticosteroids and broad-spectrum antibiotics like Azithromycin, clarithromycin to address inflammation [117,118]. Table 1 summarizes the mechanisms of action and beneficial as well as adverse effects of drug treatments used for COVID-19. 

### 6.1. RNA-Dependent RNA Polymerase Inhibitor 

#### Remdesivir

The antiviral drug, Remdesivir (VEKLURY, GS-5734), initially developed for Ebola, inhibits RNA-dependent RNA polymerase and prematurely terminates the viral RNA transcription and shows broad-spectrum antiviral activity against RNA viruses, including SARS-CoV-2 in vitro, and inhibits MERS-CoV, SARS-CoV-1, and SARS-CoV-2 replication in animal models [119]. Remdesivir is a substrate for the drug metabolizing enzymes CYP2C8, CYP2D6, and CYP3A4, as well as a substrate for organic anion transporting polypeptides 1B1 (OATP1B1) and P-glycoprotein (P-gp) transporters. Remdesivir (100–200 mg/day for 10 days) either treated alone or in combination with anti-inflammatory drugs was effective in curbing the virus and shortening the recovery time of patients undergoing treatment for COVID-19 [120].

A multicenter randomized, double-blind, clinical trial, involving 237 patients with severe COVID-19, conducted in ten hospitals in Wuhan, China, reported that seriously ill patients, receiving Remdesivir (200 mg on day 1 followed by 100 mg on days 2–10 in single daily infusions) within 10 days of symptom onset, showed a numerically faster time to clinical improvement than those receiving placebo, without any antiviral effect [121]. The study also reported early termination of the treatment due to multiple adverse events (including gastrointestinal symptoms, aminotransferase or bilirubin increases, and worsened cardiopulmonary status) in the Remdesivir-treated patients (66%) (Table 1). In a small pilot study of four critically ill COVID-19 patients with Remdesivir, three patients tested negative for SARS-CoV-2 RNA (swap test) after 3 days of therapy. However, these reports also indicated some adverse side effects including liver injury [122,123]. 

### 6.2. Viral Protease Inhibitors

#### Lopinavir-Ritonavir

Antiviral drugs such as Lopinavir-Ritonavir (Mylan or Kaletra; 400 mg and 100 mg, respectively, twice a day for 14 days), HIV protease inhibitors, used in the clinical trial provided only a moderate benefit of reducing the recovery time by 1 day [124]. Although in vivo animal study shows that a combination of Remdesivir with Lopinavir-Ritonavir yields better outcome for coronavirus infection [125]. However, the treatment with these protease inhibitors (Lopinavir-Ritonavir) develop cardio-metabolic complications including development of dyslipidemia with an adverse cholesterol profile, which could elicit inflammation with elevated reactive oxygen species (ROS) production, altered myocardial ubiquitin proteasome and calcium-handling pathways together with decreased contractile function [126,127] (Table 1). Lopinavir-Ritonavir treatment inhibits the myocardial UPS (ubiquitin proteasome system) and leads to elevated calcineurin and connexin 43 expression that may contribute to cardiac contractile dysfunction [127]. Without any benefit, Lopinavir-Ritonavir may also cause bradycardia, QT and PR interval prolongation due to the interaction with cytochrome P450 enzymes [124,128,129].

### 6.3. Attenuating Inflammation

Baricitinib (Olumiant®), an inhibitor of Janus kinase (JAK1 and JAK2) molecule and a drug for the treatment of rheumatoid arthritis was tested (2 mg or 4 mg once daily) in COVID-19 patients [130]. This drug was repurposed in COVID-19 treatment to curb the occurrence of inflammation process due to the use of ACE inhibitors, which moderately reduced the lung inflammation and cytokine [131]. The management of hyperinflammation or cytokine storm has been challenging and accounts for the majority of the mortality associated with adverse cases of COVID-19 patients. Clinical practices to address this complication involves treatment with monoclonal antibody against interleukin-6 receptor (IL-6R) such as Tocilizumab (Actrema®), Siltuximab (Sylvant®) and Sarilumab (Kevzara®) to control the infiltration of macrophages and cytokines in the respiratory system and suppression T-cell activation [132,133]. Tocilizumab specifically binds membrane-bound (mIL-6R) and soluble interleukin-6 receptor (sIL-6R) and inhibits signal transduction. COVID-19 patients treated with Tocilizumab (4 to 8 mg/kg with recommended dose of 400 mg with a maximum dose of 800 mg) in addition to routine therapy showed significant improvement of the clinical outcomes, effectively controlled body temperature with improvement of peripheral oxygen saturation and reduction of inflammatory storm [134].

Considering the emergency to identify a drug that is effective in reducing the complications associated with COVID-19, efforts are also underway to repurpose old drugs that are proven to be clinically safe. Data from RECOVERY trial indicates that dexamethasone, a steroid drug generally used as an anti-inflammatory agent, is effective in reducing the mortality rate by one-third in COVID-19 patients subjected to mechanical ventilation or who were on ventilators compared to patients receiving standard therapy [135]. Among 6400 registered COVID-19 patients, 2100 of them who received 6 mg of dexamethasone for 10 days, had reduced mortality by 20% compared to 4300 patients who were on standard treatment. More importantly, patients on ventilator support during the critical stage of treatment responded better to dexamethasone compared to patients just receiving oxygen therapy. The outcome of this study is considered a breakthrough in the fight against COVID-19 because dexamethasone is a commonly available drug and cost effective. However, further evidence is required to use dexamethasone in COVID-19 patients.

### 6.4. Hydroxychloroquine

Another drug that gained much attention for the treatment of COVID-19 is hydroxychloroquine (HCQ, Plaquenil), an anti-malarial compound, which is also widely used for attenuation of systemic lupus erythematosus (SLE), rheumatoid arthritis (RA), juvenile idiopathic arthritis (JIA) and Sjogren’s syndrome [136]. Several clinical studies, including trials from NIH (NCT04358068), are testing this drug for COVID-19 treatment, either alone or in combination with Azithromycin [137]. The treatments with hydroxychloroquine alone (400 mg by mouth twice daily for 1 day followed by 200 mg by mouth twice daily for 4 days) or in combination with azithromycin (500 mg by mouth or intravenous daily for 5 days) lead to a prolongation of the QT interval, possibly increasing the risk of sudden cardiac death [137] (Table 1). Another retrospective multicenter cohort study was conducted involving 1438 patients admitted across various hospitals in the city of New York who were diagnosed with COVID-19 (between 15–28 March 2020), those receiving either HCQ alone (dose ranges: 200–600 mg; once or twice a day) or in combination with Azithromycin (dose ranges: 200 mg to 500 mg; once or twice a day) or Azithromycin alone. The results from the study showed that the probability of death for patients receiving HCQ + Azithromycin was 25.7% (189 out of 735), while patients receiving HCQ alone was 19.9% (54 out of 271) and 10.0% (21 out of 211) in Azithromycin alone group. Cardiac arrest was significantly high in patients receiving HCQ + Azithromycin combination than treatment with placebo or HCQ alone [138]. Another cohort study performed at an academic tertiary care center in Boston, Massachusetts, showed similar high risk of QT prolongation with subsequently developed other ventricular arrhythmias in the HCQ alone (400 mg, twice on day 1, then 400 mg daily on days 2 through 5) or with Azithromycin-treated patients with COVID-19 [139]. An observational study of 1446 admitted patients to the hospital with COVID-19 (between 7 March and 8 April 2020) in New York, revealed that HCQ administration alone was not associated with either a greatly lowered or an increased risk of the composite end point of intubation or death [140]. The treatment regimen of hydroxychloroquine was a loading dose of 600 mg twice on day 1, followed by 400 mg daily for 4 additional days. However, recently, the U.S. Food and Drug Administration (FDA) revoked its approval to use HCQ for COVID-19 treatment due to disappointing results [141]. Data from randomized clinical trials suggest that HCQ had no beneficial effects compared to placebo and was not successful in decreasing the mortality rate or in hospital stay (Based on FDA report, updated on 1 July 2020) [142]. Therefore, rigorous, and large-scale studies with careful risk assessment of HCQ should be conducted prior to initiating COVID-19 therapeutics, with close monitoring cardiac manifestations including evaluation of cardiac biomarkers, routine electrocardiograms and electrolyte monitoring.

### 6.5. Vaccine Development 

There is an urgency for the development of a safe and effective vaccine for COVID-19; however, no specific vaccines against SAR-CoV-2 are currently available [40]. Multiple inactivated vaccine candidates for SARS-CoV-2, such as DNA-, RNA-based formulations, recombinant-subunits containing viral epitopes, adenovirus-based vectors and purified inactivated virus are under development [143,144]. Several candidate vaccines are still in the preliminary stage of Phase I clinical trial. The mRNA-based vaccine prepared by the USA National Institute of Allergy and Infectious Diseases against SARS-CoV-2 is under Phase 1 trial [145]. INO-4800, a DNA-based vaccine, is also in pipeline and will soon be available for human trial. Preliminary results from pilot studies and clinical trials on new vaccine are encouraging and gives hope for a successful availability of an effective vaccine by end of 2020. Several pharmaceutical companies, including Pfizer, Novartis and AstraZeneca and Moderna, are testing their candidate vaccine. University of Oxford in collaboration with AstraZeneca are in the development of COVID-19 vaccine and expect to produce 30 million doses in UK by September 2020. Jenner Institute, Oxford, UK is a leader in this effort and launched a Phase III clinical trial of more than 6000 people in May. However, due to the suspected adverse event in a person receiving the vaccine in the United Kingdom, the clinical trials have been temporarily paused.

Moderna, a USA-based company in collaboration with Switzerland’s Lonza, released positive outcomes from its Phase I clinical trial of their mRNA1273 vaccine for SARS-CoV-2 [146]. Preliminary results are very promising, showing good immune response, and due to effectiveness and safety profiles, this vaccine is approved by the U.S. Food and Drug Administration (FDA) for Phase II and Phase III studies [147,148]. Novartis announced its plans to initiate a Phase III clinical trial to study effects of canakinumab, an interleukin (IL)-1β blocker, in COVID-19 patients with pneumonia [149]. They aim to rapidly enroll 450 patients at multiple medical centers across France, Germany, Italy, Spain, UK and the USA and randomize them to receive either canakinumab or placebo on top of standard of care (SoC) [150]. Pfizer, in partnership with BIONTECH (BNT), has initiated its Phase I/II clinical trial in the USA for its mRNA-based vaccine, the BNT162 prevent COVID-19 [151]. Sinopharm, a Wuhan, China-based pharmaceutical company received approval from the National Medical Products Administration (China) and conducting Phase II clinical trials for its inactivated vaccine BBIBP-CorV. The company already tested 2000 doses of this vaccine and expect to release in the marker by the end of the year 2020. Sinovac is planning to enter its Phase III clinical trial in collaboration with Instituto Butantan in Brazil after observing positive results in its preclinical trail with the vaccine CoronaVac [144]. Ad5-nCoV, an adenovirus type 5 vector-based vaccine developed by Cansiobiologics, China is also in Phase III clinical trial and demonstrated promising effects in the early phase of testing on 108 participants [152]. Inovio pharmaceuticals in collaboration with University of Pennsylvania and Center for Pharmaceutical Research, Kansas City, Missouri, is testing its DNA-based vaccine INO-4800 [153]. Preclinical experiments conducted in guinea pigs showed antibody titer against ACE2 receptor/SARS-Cov2 binding protein. 

When countries all over the world are racing to develop their own vaccine against COVID-19, Russia has already approved a vaccine candidate for public use named Sputnik V, that was developed in collaboration with Gamaleya Research Institute of Epidemiology and Microbiology in Moscow [154]. The vaccines comprise either recombinant adenovirus type 26 (rAd26) or recombinant adenovirus type 5 (rAd5) vectors, which contain the gene for SARS-CoV-2 spike glycoprotein (rAd26-S and rAd5-S). Initial results from the ongoing Phase I and II clinical trials are promising, which include total population size of 76 healthy adult volunteers [155]. Among them 38 volunteers were intramuscularly vaccinated with Gam-COVID-Vac Lyo (lyophilized vaccine formulation) and other 38 participants were subjected to Gam-COVID-Vac (frozen vaccine formulation) [155]. Both heterologous recombinant adenoviral (rAd26 and rAd5) vector-based COVID-19 vaccines induced a strong humoral and cellular immune responses with reported safety profiles in participants. However, further investigations with larger scale population (including different underlying medical complications) are needed to demonstrate the effectiveness of this vaccine for prevention of COVID-19. Nevertheless, scientists globally have serious concerns about unforeseen adverse effects of this vaccine without the outcomes of the Phase III trial. Even though, for the development of an efficient vaccine for COVID-19, extensive preclinical studies and clinical trials are essential to carefully evaluate the adverse effect of vaccine, the aforementioned fast-paced preclinical data are encouraging for advancing the preventive strategies against COVID-19.

### 6.6. Antibody Therapy

Several other treatment options such as convalescent plasma therapy (CPT) and monoclonal antibody therapy have been evaluated with some moderate success. CPT is a traditional method where plasma containing the antibody from recovered patients infected with COVID-19 was transfused to the severely ill COVID-19 patients [25,156,157,158]. Studies showed that CP therapy was effective, and the level of neutralizing increased as high as 1:640 times in patients infected with SARS-CoV-2 [25]. Transfusing antibodies from COVID-19 survivors into high-risk patients to neutralize SARS-CoV-2 could provide a quick treatment option until an optimistic vaccine will arrive to prevent this viral infection. Efforts are also underway to design a monoclonal antibody that can target the specific epitope on the spike protein of SARS-CoV-2 and block the virus entry in to the host cell [108,159,160]. Such efforts are still in their preliminary stage [161] and are time consuming; however, they could provide a long-lasting solution for dealing with SARS viruses in general.

### 6.7. Stem Cell Therapy

Recently, stem cell therapies with secreted extracellular vesicles (EVs) offer a potential therapeutic benefit in COVID-19 patients by attenuating inflammation with regeneration of the damaged lung. Mesenchymal stem cells (MSCs)-derived EVs-based therapy could be the most promising reparative strategy in people with COVID-19, because of its high proliferation rate, low invasive nature, and the immunomodulatory, antioxidant and anti-inflammatory properties of MSCs [162]. There are several promising clinical trials with MSC-derived EVs underway, which could reveal convincing evidence in the encouraging prospect of MSC-based therapies for respiratory complications of COVID-19 patients [163,164]. 

Despite the above-mentioned beneficial effects of different therapeutics, the safety profiles of these therapies have not been proficiently identified. Specifically, the potential adverse cardiovascular effects of these drugs in COVID-19 patients need urgent attention before rushing the approval of any new drug into clinical application. For most effective treatments for COVID-19, it is important to pay attention to emerging evidence about potential harmful risk of drug interactions.

## 7. Conclusions

Due to the highly transmissible novel coronavirus, SARS-CoV-2, the COVID-19 outbreak has become an unprecedented worldwide pandemic with a record number of infected individuals and an excess of mortality. The desperate need for effective therapeutics for COVID-19 during this pandemic integrated scientist around the world across multiple research fields while sharing their research findings and knowledge to fast-track the process of drug discovery.

Considering the high mortality of COVID-19 patients with cardiovascular comorbidities, it is important to understand whether it is attributable to underlying cardiovascular disease (CVD) or if CVD is the consequence of inflammatory response to SAR-CoV-2 infection or severe respiratory symptoms. The precise mechanisms linking CVDs and worsened prognosis or higher mortality rate in COVID-19 patients remain unknown. Recent therapies under investigation for severe multi-organ failure in COVID-19 patients may have adverse cardiovascular effects, while their clinical efficacy for combating COVID-19 is yet to be established. New advanced technological tools, like information technology based on smart phone apps, social media, artificial intelligence (AI), machine learning, etc., accelerate the diagnosis/screening of patients with virus, analysis of available literature, and identification of potential therapeutic targets and other specific clinical features to tackle COVID-19 pandemic. Moreover, AI, particularly, plays an important role in predicting the harmful interaction between cardiovascular consequences with the drugs used for COVID-19, by automated interpretation of collected meta-data from various sources. 

In the context of disease progression with cardiovascular complications, the researchers are focusing on developing new drugs in parallel to repurposing already clinically approved drugs to avoid a massive surge of COVID-19 patients with a prevalence of CVD. Therefore, urgent understanding of molecular mechanism as well as retrospective and prospective studies with robust diagnosis of cardiovascular impairments will be crucial for development of advanced therapies for the treatment of SARS-CoV-2 virus, which could mitigate the adverse cardiovascular events among COVID-19 patients and save humankind around the globe from this deadly pandemic.

## Figures and Tables

**Figure 1 ijms-21-06790-f001:**
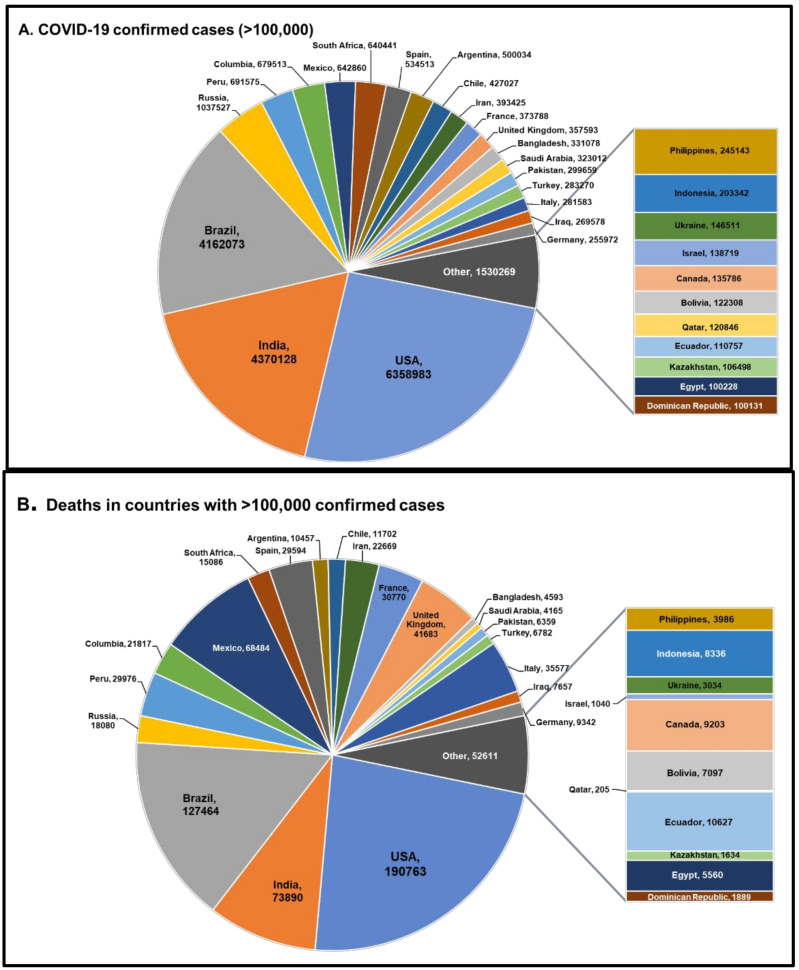
Demographics of COVID-19 cases and deaths. (**A**) COVID-19 confirmed cases by countries (>100,000 cases) and (**B**) global deaths. (**C**) COVID-19 confirmed cases by countries (>50,000 cases and < 100,000) and (**D**) global deaths. Based on data obtained from COVID-19 dashboard by Center for Systems Science and Engineering (CSSE) at Johns Hopkins University (JHU).

**Figure 2 ijms-21-06790-f002:**
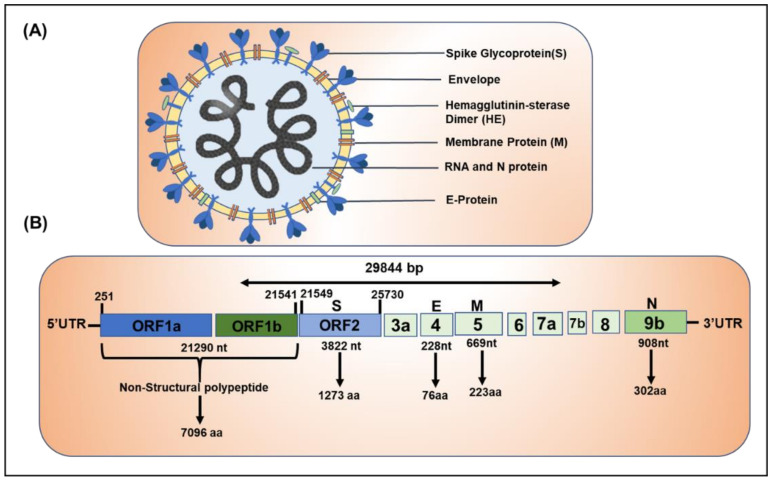
Typical ultra-structure and genome organization of SARS-CoV-2. (**A**) Illustration of SARS-CoV-2 showing key components of the virus morphology. The virus contains several crown or club shaped glycoprotein spikes (S) on the outer membrane used for anchoring to host cell, lipid bilayer spanning membrane protein (M), hemagglutinin esterase (HE) in the envelope, which has enzymatic activity that weakens the host defense, envelope (E) and the genetic material RNA and nucleoprotein (N). (**B**) Genomic organization of novel SARS-CoV-2. The genome of SAR-CoV-2 is approximately 2.9 kb in size. The first ORF1 (2.1 kb) codes for two (Frameshift) non-structural polypeptide pp1a and pp1b (weighs-7.09 KD) and several non-structural proteins (nsps). Four major structural proteins are coded from different ORFs (1) spike (S), (2) envelope (E), (3) membrane (M), and (4) nucleocapsid (N).

**Figure 3 ijms-21-06790-f003:**
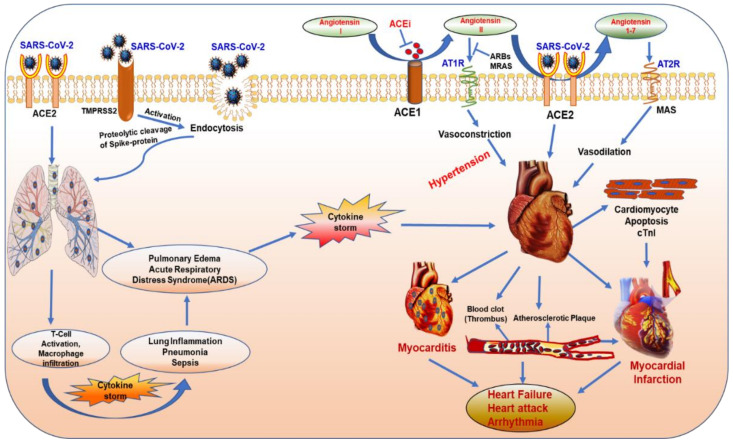
Pulmonary and cardiovascular complications associated with COVID-19 infection. The abundance of SARS-CoV-2 compromises the normal function and leads to complications in lungs (inflammation, hypoxia, cytokine storm, pulmonary edema, acute respiratory distress syndrome) and in heart (myocardial infarction, heart failure, myocarditis and arrhythmia). ACE1, angiotensin I-converting enzyme; ACE2, angiotensin-converting enzyme 2; ACEi, ACE inhibitor; AT1R, angiotensin type 1 receptor; AT2R, angiotensin type 2 receptor; ARBs, angiotensin II type-I receptor blockers; cTnI, cardiac troponin I; MAS, mitochondrial assembly receptor; MRAs, mineralocorticoid receptor antagonists; TMPRSS2, transmembrane serine protease 2.

**Figure 4 ijms-21-06790-f004:**
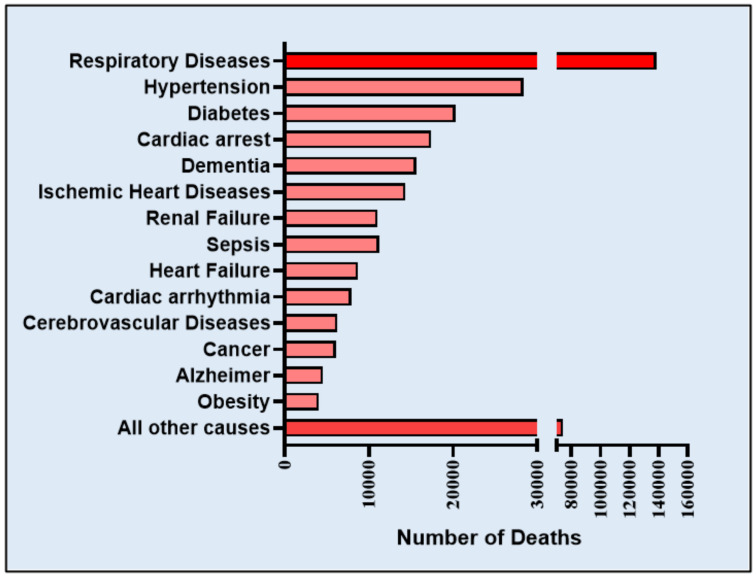
Conditions contributing to the deaths of COVID-19 patients in the United States of America (according to Death Data of CDC as of 25 July 2020). Deaths involving more than one condition (e.g., deaths involving both diabetes and respiratory arrest) were counted in both totals.

**Table 1 ijms-21-06790-t001:** Mechanisms of action and adverse cardiac effects of drug treatments for COVID-19.

Drug	Mechanism of Action	Drug Interaction	Side Effects	Cardiac Adverse Events
Remdesivir (GS-5734) **Antiviral drug**	Inhibits RNA-dependent RNA polymerase. Terminates the viral RNA transcription.	Hydroxychloroquine or chloroquine	Increases levels of liver enzymes, liver inflammation and liver damage. Reduces antiviral activity when co-administered with chloroquine or hydroxychloroquine	Lowers blood pressure and results in cardiopulmonary failure [122,123]
Lopinavir- Ritonavir **Antiviral drug**	Inhibits protease enzyme activity in infected cells and reduces virus replication. Strong inhibitors of CYP3A4.	Antiarrhythmic, Anticoagulant, Antiplatelet, Statins	Hepatoxicity, dyslipidemia	Inhibits myocardial UPS (ubiquitin proteasome system) and leads to elevated calcineurin and connexin 43 expression that may contribute to cardiac contractile dysfunction [126,127]. Cardio-metabolic complications. Bradycardia, QT and PR interval prolongation due to the interaction with cytochrome P450 enzymes [124,128,129].
Baricitinib **Protein Kinase inhibitor**	Inhibitor of Janus kinase (JAK1 and JAK2) and mainly used treatment of severe rheumatoid arthritis	Lopinavir or ritonavir and remdesivir	Unknown	Unknown
Tocilizumab/ Sarilumab **Humanized monoclonal antibody against IL-6-**	Binds to membrane-bound (mIL-6R) and soluble interleukin-6 receptor (sIL-6R) and inhibits signal transduction and used as immune-suppressive drug to treat severe rheumatoid arthritis	Unknown	Drug interaction may impact inflammation and impairment of drug metabolism. Hypercholesterolemia	Hypertension
Siltuximab **Chimeric monoclonal antibody anti-IL-6**	Blocks the activation of IL-6 mediated inflammation and mainly used in neoplastic cancer.	Unknown	Unknown	Unknown
Hydroxychloroquine (HCQ) **Antiviral drug**	Increases lysosomal pH in antigen-presenting cells and blocks toll-like receptors during inflammation on plasmacytoid dendritic cells (PDCs). Primarily used as antimalarial drug and in systemic lupus erythematosus and rheumatoid arthritis	Azithromycin or Lopinavir-Ritonavir or Antiarrhythmic QT-prolonging agents	Showed no beneficial effects compared to placebo and could not decrease the mortality rate or hospital stay.	Results in QT prolongation, QRS prolongation, bradycardia and tachycardia with increase the risk of sudden cardiac death [138]. Ventricular arrhythmia, fibrillation and tachycardia [137,138,139]
Mesenchymal stem cells (MSCs)-derived extracellular vesicles (EVs)	Compete with virus for cellular uptake. EV could contain the small interfering RNA (siRNA) to interrupt the virus activity	Unknown	Unknown	Unknown

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
