# Peer review of "Cardiovascular Complications Associated with COVID-19 and Potential Therapeutic Strategies"

_ijms, 2020, doi:10.3390/ijms21186790_

Round 1
Reviewer 1 Report
The authors summarized the current progress of COVID-19, which has been associated with multiple organ dysfunction syndrome, especially including severe cardiac impairment. The authors showed the significant evidences, revealing a direct correlation between COVID-19 and cardiovascular complications. This manuscript also summarized current treatment on COVID-19, and highlight the importance of integrated approach with cardiovascular syndromes.
It could be better to summarized the molecular mechanisms regarding how COVID-19 impaired cardiomyocytes.
Author Response
Thank you for the encouraging comments.
We greatly appreciate the reviewer’s suggestions to summarize the molecular mechanisms of cardiomyocyte injury due to COVID-19 infection. Although, the exact molecular mechanism by which SARS-COV-2 virus leads to cardiomyocyte injury is not identified, however, we incorporated three paragraphs (in blue color) regarding the potential mechanisms associated with SARS-CoV-2 infection, which could injure cardiomyocytes with adverse cardiac events (in page 10- lines 371-411).
Reviewer 2 Report
The paper has been improved.
In my opinion the manuscript is now acceptable for pubblication in IJMS
Author Response
Reviewer#2:
The paper has been improved.
In my opinion, the manuscript is now acceptable for publication in IJMS
We greatly appreciate the reviewer’s inspiring remarks.
We have made a considerable effort to take into account the suggestions proposed by all the reviewers. Hope the revised manuscript could satisfy the reviewer’s concern.
This manuscript is a resubmission of an earlier submission. The following is a list of the peer review reports and author responses from that submission.
Round 1
Reviewer 1 Report
This is a relevant and timely review regarding the cardiovascular complications in COVID-19 patients. However, I would like to address some issues that if resolved I believe would improve this paper.
- The introductory part (subsections 1 through 3) on virus biology, structure, genomic, and organization is redundant. There are numerous, if not hundreds of papers that go in every detail on this virus, its biology, replication cycle, host entry, etc. Please stick to a couple of main and most robust papers in this area, preserve the image but significantly reduce this portion of the text and focus on CV-related complications as this is the point of the review
- A major shortcoming is the somewhat erratic organization of this manuscript. From all CV complications for some reason, only myocarditis is elaborated and then authors went on explaining diabetes mellitus as a risk factor for increased mortality in COVID-19. Authors do not cover important components of CV complications such as myocardial infarction/ACS, heart failure in these patients, and also, very importantly, thromboembolic aspect of COVID-19. I would recommend discarding DM from this analysis and to be focused exclusively on CV abnormalities and complications
- This manuscript requires significant revision and introduction of new subsections dealing with heart failure, myocardial infarction, myocardial injury, and thromboembolic risk/hypercoagulability, and microvascular thrombosis in these patients which are all very well established. On the other hand, general portions of the virus biology, etc. and other risk factors, other than CV-oriented ones should be kept to the minimum.
- An appropriate graphic depicting CV complications would be welcome to accompany this manuscript.
Reviewer 2 Report
The authors summarized the mechanisms of coronavirus disease 2019 (COVID-19) and its association with cardiovascular complications and diabetes mellitus. The information in this manuscript is the most updated, and introduces the current methods for treatment of COVID-19, and ongoing efforts for discovery of drugs and vaccines. The review provides important information, which will be considered for minimizing the cardiovascular risk among COVID-19.
Here is a minor comment:
Please show figure 2, which is missing in the manuscript.
Reviewer 3 Report
This work does not add new informations to the literature
Author Response
Responses to Reviewer#3
This work does not add new informations to the literature
Responses: In this review article, we summarized the evolving research studies pertaining to cardiovascular complication related to COVID-19 during this worldwide unprecedented pandemic. In the revised article, we incorporated the up-to-date several case reports with cardiovascular syndromes related to COVID-19, specifically, hypertension, myocardial injury (including myocardial infarction, heart failure, coagulopathy, and thromboembolism), myocarditis and myocardial arrhythmias under separate subheading. We also discussed about the number of promising therapies under investigation to treat and prevent COVID-19.
We have made a considerable effort to take into account the suggestions proposed by all the reviewers. Hope the revised manuscript could satisfy the reviewer’s concern.
Round 2
Reviewer 1 Report
The authors have addressed all of my concerns.
Author Response
Thank you for your valuable comments
Reviewer 3 Report
The Authors have summarized a lot of informations on this pathological condition but there is no critical contribute to improve The Diagnosis and The treatment
Author Response
The Authors have summarized a lot of informations on this pathological condition but there is no critical contribute to improve The Diagnosis and The treatment
Response: We have included a new paragraph to discuss about the diagnosis of COVID -19 in “6. Treatment Options and Vaccine Development”. We also explained in detail the consequences of treatment with ACE inhibitor to improve our manuscript.
